# Composite Lamina Model Design with the Use of Heuristic Optimization

**DOI:** 10.3390/ma16020495

**Published:** 2023-01-04

**Authors:** Artem Balashov, Anna Burduk, Jozef Husár

**Affiliations:** 1Department of Laser Technologies, Automatization and Product Organization, Faculty of Mechanical Engineering, Wroclaw University of Science and Technology, 50-370 Wrocław, Poland; 2Department of Industrial Engineering and Informatics, Faculty of Manufacturing Technologies with a Seat in Prešov, Technical University of Košice, Bayerova 1, 080-01 Prešov, Slovakia

**Keywords:** heuristic optimization, tabu search, composite material

## Abstract

In engineering practice, a problem is quite often faced in which the number of unknown parameters exceeds the number of conditions or requirements or, otherwise, there are too many requirements for too few parameters to design. Such under- or over-defined tasks are sometimes not possible to solve using a direct approach. The number of solutions to such problems is multiple, and it is most rational to search for the optimal one by numerical methods since the more unknown design parameters there are to be designed, the more potential solutions there are. This article discusses a way to find an optimal solution to such an underdetermined problem by heuristic optimization methods on the basis of the example of designing a composite wing skin of an aircraft. Several heuristic approaches, specifically gradient descent and Tabu search, are studied to solve the design problem and to locate an optimal solution. They are also compared to a conventional direct approach. The examined composite lamina is optimized by the target function of minimum weight with the constraints of strength and buckling failure criteria. In most of the observed cases, the heuristic method designed structures which were considerably better than the structures that were obtained by conventional direct approaches in terms of the weight to load ratio.

## 1. Introduction

In the beginning, a short definition of the problem will be provided. During the process of designing a part or a structure, it is the goal to define the parameters that are not yet known in a way to satisfy the operation requirements, such as conditions of strength, buckling, rigidity, or other types of destructibility or unsuitability for the operation of the structure. Conventionally, we try to equate the number of unknown parameters to the boundary conditions so the problem will be fully defined and will have only one solution that can be determined by using analytical methods, such as solving the system of linear equations or similar. In other words, we try to define each single unknown parameter with its own requirement, sometimes within interdependence, yet always coinciding [1]. 

However, it is not always possible to equalize the number of conditions with the designed parameters; too many requirements of different operation conditions could be applied to a design structure so the task would become redefined. What is more, too many properties of the final structure must be designed so that the number of feasible solutions would make a task underdefined. Such cases happen more often with the design of structures that are made from composite materials, due to their anisotropy and overall layered structure [2].

The number of possible solutions in these situations could become excessive enough to make the task not solvable with a direct conventional approach. The only possible way to solve these problems, other than introducing some assumptions which would limit possible solutions by any means, is to use numerical optimization. Specifically, a heuristic approach is suggested in this article. It will, by definitions, not provide the only best solution but could enhance finding an optimal solution to an otherwise unsolvable problem [3].

There is no single superior optimization approach. The survey of the current continuous nonlinear multi-objective optimization concepts and methods is well presented in article [4]. A comprehensive review of the recent developments in the computational modelling of plates and shells, including over 800 relevant references, is presented in article [5]. Sufficient results using Genetic algorithms that were applied for the pre-buckling and buckling analysis of general variable stiffness plates were made in Wu et al. [6]. In article [7], the authors propose a modified genetic algorithm to design a wing turbine blade within multi-objective optimization.

### 1.1. An Observed Case Definition

This article discusses the construction of a composite wing skin of an aircraft. This task, although at different levels, is both an underdefined and redefined engineering task. Numerous operational, technological, and production characteristics had to be neglected in this article since the example is an introductory one.

The problem considers the design of a composite wing skin according to the conditions of strength (tensile, compression, and shear) and buckling. The number of projected parameters significantly exceeds the number of conditions. These are: the material (or materials) of the reinforcing component, the number of layers, and the orientation of each layer which make the task globally undetermined. At the same time, several calculated cases (the state under consideration or operation condition) will be considered for each group of conditions, making the local subtasks redefined [8].

### 1.2. Assumptions

Only aerodynamic environmental forces (lifting force and drag force) have an influence on the wing skin (Figure 1). The lifting force ensures the carrying of the fuselage with passengers or cargo in the air, and the air resistance is overcome by the thrust of the engines. Being distributed over the span of the wing, the lifting force can be transferred to the fuselage to balance the gravitational forces by means of a structural element working on bending. Such elements include beams, shells, and plates. The principle of their operation is nearly the same and similar to a beam or a thin-walled rod [9].

The distributed external load is transformed into a system of normal and tangential stresses. Hence, the skin panel is subjected to the forces in the plane and distributed surface pressure (Figure 2).

With rare exceptions, the skin is made with a symmetrical structure, i.e., from an orthotropic composite in the x-y axes. Both orthotropy and symmetry of laying plies in thickness are fully justified, since in this case, during force and temperature loading, undesirable deformations and stresses do not occur in the panel. They are associated, for example, with mutual influence coefficients when normal stresses cause shear deformations. If these shear deformations are constrained by the frame, then additional tangential stresses appear [3]. An imbalance of the structure in thickness (the asymmetry of the layer package) when the temperature changes, leads to the appearance of temperature moments (bending and torsional), which cause warping of the panel [8].

Strength characteristics, since they are considered in the plane of the plate, do not depend on its dimensions, but on the characteristics of the loss of stability of the plate during compression or torsion. For this reason, we will consider the wing skin in certain theoretical parts (Figure 3), limited in the longitudinal direction by longeron or longitudinal walls, and in the transverse direction by ribs [9].

The present article follows the structure below.

The “Introduction” chapter describes the necessity of the followed research and the innovation of the proposed approach. The problem definition is described in first subchapter “an observed case definition”, and more detailed explanations and simplifications to the observed case are given in the subchapter “Assumptions”. They are needed in order to provide the information that is required only for this study and to introduce the aircraft wing skin characteristics.

The section “Materials and Methods” introduces a more in-depth description of the case study. The justification of multiple observe scenarios is provided. A short overview of Kirchoff hypotheses and classical theory of thin plates is provided in the subchapter “Failure criteria of strength and buckling”. The subchapters “Strength failure criterion” and “Buckling failure criterion” provide the required equations. The subchapter “Case study” describes an example of conventional computation algorithms for thin composite laminas. It considers two failure criteria and several loading scenarios.

The section “Results” describes the results that were obtained by using heuristic optimization in the observed case study. The proposed approach is introduced and reported in the subsection “Greedy neighborhood”. The methodology verification is provided at the end of the “Result” section by providing computation and intercomparison with a conventional direct algorithm for 10 more typical cases.

The importance and innovation of the proposed approach is provided in the section “Conclusion”.

## 2. Materials and Methods

All the structures operate in different conditions and, depending on this, they are affected by different natures and magnitudes of the load. Naturally, the product must always remain functional. Depending on the maneuver that is performed, the wing of the aircraft bends up or down (Figure 4), then the upper (or lower) skin is stretched in one case and compressed in the other. These scenarios are most typical for an aircraft to encounter during a stable flight (Figure 4, left) when the lifting force acts vertically upwards, and when the aircraft is stationed in a hangar (Figure 4, right) with the gravity forces acting on the wing vertically downwards.

This type of redefined tasks implies several different conditions in one group of conditions. The simplest way to find an optimal solution in such a situation is shown in Figure 5 (from the next chapter). It consists of finding the dependencies of the desired parameter on the objective function, taking into account the constraints for each condition, and then choosing the optimal solution that satisfies all the calculated cases [10].

In the simplest case that is applicable to the considered problem, the method consists of finding the dependence of the maximum specific strength of the layer on its orientation for the given material under load. For each case of loading, which are revised, after which the graphs are plotted in one coordinate system, the envelope curve of all the graphs is plotted. The lower point of this envelope curve will be the optimal orientation for all the considered loading cases [11].

Subsequently, this operation is performed in one form or another in order to find the orientation of the layer to ensure that the stability condition is met, for each layer specifically.

The advantage of this analytical design methodology is that it is able to find a superlative solution that is the most optimal one. The disadvantage is the need to conduct a full search for all possible solutions.

### 2.1. Failure Criteria of Strength and Buckling

As it is known from structural mechanics, the generalized linear shearing forces Q_x_, Q_y_; bending moments M_x_, M_y_; and torque M_xy_ as well as normal (axial) N_x_, N_y_, and tangential q_xy_ forces act at each point of the plate.

The classical theory of thin plates with a symmetrical arrangement of layers, based on the Kirchhoff hypotheses [12], describes the stress-strain state of the plate.

Further, in Figure 5, a schematic diagram of the composite panel structure is shown for a better understanding. x and y are the axes of lamina itself and 1i and 2i are the axes of each layer i. N_x_, N_y_, and q_xy_ are the linear and shear loads acting in the plane of lamina, σ_1i_, σ_2i_, and τ_12i_ are the stresses acting on each layer, respectively, along, across, and tangential. φ is the orientation angle of the specific layer.

Further in this article, a case study of designing the composite lamina and its heuristic optimization will be described. The failure criteria of strength and bucking will be used as optimization constraints as these criteria are most commonly obligatory to fulfil. 

### 2.2. Strength Failure Criterion

There are many criteria for the destruction of the material. However, in this paper the criterion of destruction of the Mises–Hill is considered. For the layered composite plates, it takes the form (Equation (1)) as follows:(1)$$\Phi_{\mathrm{st}}=\frac{\sigma_{1i}^{2}}{F_{1i}^{2}}-\frac{\sigma_{1i}\sigma_{2i}}{F_{1i}F_{2i}}+\frac{\sigma_{2i}^{2}}{F_{2i}^{2}}+\frac{\tau_{12i}^{2}}{F_{12i}^{2}}\;\leq 1$$
where F_1_, F_2_, and F_12_ in this formula are the strength limits along and across the fiber and shear, respectively. They are determined according to the stresses from the regulatory documents for materials or experiments. If the stress is positive, the tensile strength is selected for tensile, if negative it is selected for compression [13].

Here, sigma (σ) is the stress of each layer in its local coordinate system. This value is to be determined. The standard algorithm for designing the structure of the plate and determining its thickness and the orientation of the reinforcement fibers according to the strength condition is simplified below (Equations (2)–(5)):Determination of the coefficients of the layered stiffness of the package B:
(2)$$B_{11}=\overset{n}{\underset{i=1}{\sum }}\delta_{i}\left[\bar{E_{1i}}\cos^{4}(\varphi_{i})+2\bar{E_{1i}}\mu_{21i}\sin^{2}\varphi_{i}\cos^{2}(\varphi_{i})+\bar{E_{2i}}\sin^{4}(\varphi_{i})+G_{12i}\sin^{2}\left(2\varphi_{i}\right)\right]$$
(3)$$B_{12}=\overset{n}{\underset{i=1}{\sum }}\delta_{i}\left[\left(\bar{E_{1i}}+\bar{E_{2i}}\right)\sin^{2}(\varphi_{i})\cos^{2}(\varphi_{i})+\bar{E_{1i}}\mu_{21i}\left(\sin^{4}(\varphi_{i})+\cos^{4}(\varphi_{i})\right)-G_{12i}\sin^{2}\left(2\varphi_{i}\right)\right]$$
(4)$$B_{22}=\overset{n}{\underset{i=1}{\sum }}\delta_{i}\left[\bar{E_{1i}}\sin^{4}(\varphi_{i})+2\bar{E_{1i}}\mu_{21i}\sin^{2}(\varphi_{i})\cos^{2}(\varphi_{i}+\bar{E_{2i}}\cos^{4i}\varphi_{i}+G_{12i}\sin^{2}\left(2\varphi_{i}\right)\right]$$
(5)$$B_{33}=\overset{n}{\underset{i=1}{\sum }}\delta_{i}\left[\left(\bar{E_{1i}}+\bar{E_{2i}}-2\bar{E_{1i}}\mu_{21}\right)\sin^{2}(\varphi_{i})\cos^{2}(\varphi_{i})+G_{12i}\cos^{2}\left(2\varphi_{i}\right)\right]$$
where $\bar{E_{1}}$ and $\bar{E_{2}}$ are adduced Young’s modulus. E_1_, E_2_, and G_12_ are Young’s modulus along and transverse fiber, and Shear modulus. µ_12_ is normal Poisson’s ratio, µ_21_ is transverse Poisson’s ratio, and δ_i_ is the thickness of layer i:
(6)$$\bar{E_{1}}=\frac{E_{1}}{1-\mu_{12}\mu_{21}};\bar{E_{2}}=\frac{E_{2}}{1-\mu_{12}\mu_{21}};\mu_{21}=\frac{E_{2}}{E_{1}}\mu_{21}$$Determination of deformations of a package of layers in the general (global) coordinate system of the entire package (Equation (7)):
(7)$$\varepsilon_{y}=\frac{N_{x}B_{22}-N_{y}B_{12}}{B_{11}B_{22}-B_{12}{}^{2}};{\;\varepsilon }_{y}=\frac{N_{y}B_{11}-N_{x}B_{12}}{B_{11}B_{22}-B_{12}{}^{2}};{\;\gamma }_{\mathrm{xy}}=\frac{q_{\mathrm{xy}}}{B_{33}}.$$
where N_x_ is the longitudinal force in the skin plane, N_y_ is the transverse force, q_xy_ is the tangential force flow, and B_11 (12, 22, 33)_ are the stiffness coefficients.Determination of deformations of each layer in its own (local) coordinate system (Equations (8)–(10)):
(8)$$\varepsilon_{1i}=\varepsilon_{x}\cos^{2}(\varphi_{i})+\varepsilon_{y}\sin^{2}(\varphi_{i})+\gamma_{\mathrm{xy}}\sin(\varphi_{i})\cos\left(\varphi_{i}\right)$$
(9)$$\varepsilon_{2i}=\varepsilon_{x}\sin^{2}(\varphi_{i})+\varepsilon_{y}\cos^{2}(\varphi_{i})-\gamma_{\mathrm{xy}}\sin(\varphi_{i})\cos(\varphi_{i})$$
(10)$$\gamma_{1i}=\left(\varepsilon_{x}-\varepsilon_{y}\right)\sin(2\varphi_{i})+\gamma_{\mathrm{xy}}\cos(2\varphi_{i})$$
Knowing the elastic characteristics of each layer from its deformations, it is possible to determine the loads (Equations (11)–(13)) that are acting in the layer according to the physical law:
(11)$$\sigma_{1i}=\bar{E_{1i}}\left(\varepsilon_{1i}+\mu_{21i}\varepsilon_{2i}\right)$$
(12)$$\sigma_{2i}=\bar{E_{2i}}\left(\varepsilon_{2i}+\mu_{12i}\varepsilon_{1i}\right)$$
(13)$$\tau_{12i}=G_{12i}\gamma_{12i}$$


The main catch in this algorithm is that the stiffness coefficients, as can be seen from the formulas, depend both on the orientation of each layer and on their quantity (or total thickness) [14]. Even for one calculation case it is difficult to rewrite the formulas to determine the dependence of the tensile strength on some specific characteristics. This can only be done for some simple structures, such as 0°, 90° or +-φi.

### 2.3. Strenght Buckling Failure Criterion

The panel buckling criterion (Equation (14)) is as follows:(14)$$-\frac{N_{x}}{N_{x\;\mathrm{crit}}}-\frac{N_{y}}{N_{y\;\mathrm{crit}}}+\frac{q_{\mathrm{xy}}^{2}}{q_{\mathrm{xy}\;\mathrm{crit}}^{2}}\leq 1$$

Here, $N_{x\;\mathrm{crit}}$, $N_{y\;\mathrm{crit}}$, and $q_{\mathrm{xy}\;\mathrm{crit}}^{2}$ are the critical loads at which the panel does not buckle.

The critical loads are determined by various methods while this study presents one of the most frequently used. In order to do this, the bending stiffness of the panel is first determined (Equations (15)–(19)):(15)$$D_{1}=2\sum_{i=1}^{n/2}b_{11i}\left[\frac{\delta_{i}^{3}}{12}+\delta_{i}{\left(\sum_{k=1}^{i}\delta_{k}-\frac{\delta_{i}}{2}\right)}^{2}\right];$$
(16)$$D_{2}=2\sum_{i=1}^{n/2}b_{22i}\left[\frac{\delta_{i}^{3}}{12}+\delta_{i}{\left(\sum_{k=1}^{i}\delta_{k}-\frac{\delta_{i}}{2}\right)}^{2}\right];$$
(17)$$D_{12}=2\sum_{i=1}^{n/2}b_{12i}\left[\frac{\delta_{i}^{3}}{12}+\delta_{i}{\left(\sum_{k=1}^{i}\delta_{k}-\frac{\delta_{i}}{2}\right)}^{2}\right];$$
(18)$$D_{33}=2\sum_{i=1}^{n/2}b_{33i}\left[\frac{\delta_{i}^{3}}{12}+\delta_{i}{\left(\sum_{k=1}^{i}\delta_{k}-\frac{\delta_{i}}{2}\right)}^{2}\right];$$
(19)$$D_{3}=D_{12}+2D_{33}$$
on which the estimated number of half-waves of buckling failure is determined.
(20)$$m\left(m-1\right)<c^{2}\sqrt{\frac{D_{2}}{D_{1}}}<m\left(m+1\right);$$
(21)$$n\left(n-1\right)<\frac{1}{c^{2}}\sqrt{\frac{D_{1}}{D_{2}}}<n\left(n+1\right);$$
where c is a/b just for the convenience of recording (Equations (20) and (21)). 

Next, the coefficients of support k_x_, k_y_, and k_xy_ are calculated according to semi-empirical formulas that take up too much space to put them in this article. However, they can be found in [15] or in reference Table [12]. In the context of this article, it is important to understand that the support coefficients depend on the bending stiffness and the number (Equations (22)–(25)) of longitudinal and transverse half-waves of the stability loss.

The functional dependences of the critical loads of the buckling failure on the structural parameters are presented below.
(22)$$N_{x\;\mathrm{crit}}=\frac{k_{x}\pi^{2}\sqrt{D_{1}D_{2}}}{\mathrm{ab}};\;$$
(23)$$N_{y\;\mathrm{crit}}=\frac{k_{y}\pi^{2}\sqrt{D_{1}D_{2}}}{\mathrm{ab}};$$
(24)$$q_{\mathrm{xy}\;\mathrm{crit}}=\frac{k_{\mathrm{xy}}\pi^{2}\sqrt{D_{1}D_{2}}}{\mathrm{ab}};\;$$

The formulas were adopted from [15,16] and modified for this particular study. Figure 3, Figure 4 and Figure 5 were also borrowed from the same references. The main nuance of using this algorithm for a panel design and determining structural parameters, is the same problem as with the previous algorithm. In order to determine the stiffness, it is necessary to already know the number of layers and their orientation. Moreover, the bending stiffness also depends on the order of the layers [17].

### 2.4. Case Study

As an example and demonstration for this article, the design of the panel structure of one theoretical section of the wing skin with specified conditions is considered.

The loads for the plate are set as in Table 1:

The plate dimensions set as 600 × 600 mm^2^.

The frequently used composite materials [12,17] and their characteristics are shown in Table 2.

The material of the layers for the first calculation is chosen first from Table 2—AS4 63%. The definition of the material is the search for another unknown. It is often done by comparing the material with others (i.e., numerical search for the optimum), or choosing a material that is already used in other designs for technological and logistical convenience (abstractly speaking, adding conditions to the levelling system to compensate for the unknowns), etc. [18].

First, the determination of the optimal reinforcement angles of the package for each design case is done, provided that the entire package is executed with a ± φ structure. After that, the construction of the envelope curve is made by finding the optimal reinforcement angle that satisfies all the calculated cases. As can be seen in the graph, the optimal orientation of the layers is ±38° degrees.

The optimum orientation of layers is understood as the structure of composite lamina which satisfies the loading condition and has the lowest weight. The weight criterion will also be the target function of optimization in the next chapter and Figure 6 and Figure 7. The minimum weight goal function is the most common for aircraft structures as it allows the significant reduction of the exploitation costs.

The next step is the determination of the optimal reinforcement angle to satisfy the buckling conditions for all the observed cases for a unidirectional ±φ structure. It is then plotted in the graph, together with the graphs of the reinforcement angles from the strength conditions. Then, the envelopes curves for each observation case are drawn and another envelope of envelopes is made [16]. The obtained optimal reinforcement angle of the package in ±27° is valid for a unidirectional structure. The thickness in this case is 6.38 mm.

There are an infinite number of partly empirical, partly numerical, partly analytical ways to try to optimize this structure. For example, it is possible to adopt a structure that provides strength conditions as the main ones, and then add layers that are optimal for providing buckling conditions, creating a structure [±38°, ±27°]. However, in this particular example, this method will not be of benefit. The number of the required added layers of ±27° to the end is 40 pieces, which makes the optimized structure the same in thickness as just ±27°.

In addition, it is worth recalling that since in the analytical algorithm there was a transformation of formulas to find the desired from the known and not an iterative numerical approach, then the result turned out to be theoretical and constructive. In addition, it does not take into account the real number of layers and the thickness of each monolayer [19]. In fact, the result should be rounded to a multiple of the number of layers, which in this case will be 6.72 mm.

## 3. Results

The main idea of this article is the application of numerical heuristic optimization methods to search for the optimal solution. Possible optimization parameters will be the orientation of the layer, its position relative to the mid-plane of the package (since bending stiffness depends on it), and its own material. The power of the problem under consideration will be n!^(180/p) k^, where p is the orientation step (since with modern production technologies it is almost impossible to observe the tolerance in +-2-3° of laying layers). It is recommended to be set in a range from 5° to 15° and k is the number of potential materials. There are globally two optimization options. First, try to optimize an existing result or find the optimal solution from the beginning. The second one is revised in this article.

### Greedy Neighborhood

The neighborhood of the solution in this group of methods will take all the possible orientations and materials of the reinforcing fiber inside one layer. That is 180/p∙K. The idea is to somehow (using one or another heuristic [20,21,22]) find the best (optimal) solution in the neighborhood, assign such a material and orientation to this layer, and proceed to the next layer. It is important to remember here that the reference frame of the layers is not from the bottom up (or from the top down) relative to the thickness of the package but from the middle plane, two layers at a time (+φi and −φi).

Firstly, a full search among all the possible solutions with a step of 15° was conducted. That is, only orientations [0°, ± 15°, ± 30°, ± 45°, ± 60°, ± 75°, 90°] were considered. The group of materials from Table 1 were selected to be epoxy-compatible. These are materials 1–7, 13, 14. The results of the full greedy search are shown in Figure 8.

Based on this graph, it is clearly observable how the algorithm searches for the entire area of the reinforcement orientation at first, then for the area of the materials. The optimal structure found in this case is: 

[[‘AGP370-5H, 62%’, ‘±60°’, ‘2 layers’], 

  [‘AGP370-5H, 62%’, ‘±45°’, ‘2 layers’], 

    [‘AGP370-5H, 62%’, ‘0°’, ‘3 layers’]]. 

By analyzing the results of the algorithm, its imperfection is understandable as it went through all the materials and orientation angles for each layer, but as a result, it decided to use only one material AGP370-5H, 62%, which, frankly, does not have the best characteristics for these requirements. This happened because this material has one monolayer that is thicker than the others and adding a layer that was not the most optimal material but several times thicker than the others was optimal from the point of view of the algorithm [23]. 

The attempt was made to correct the algorithm by forcing it to search among the materials with the specific thickness characteristics, and after finding the optimal one, adding it to the original material with the actual thickness of the monolayer to the stack (Figure 9).

Based on this graph, it can be observed that when going through all the orientation angles and materials, the thickness of the package (first the lower curve) changes discretely when adding a layer, and not when changing the material. The optimal structure that was found in this case is:

[[‘IM6, 65%’, ‘±55°’, 2];[‘IM6, 65%’, ‘±20°’, 2];[‘IM6, 65%’, ‘±70°’, 2];

 [‘IM6, 65%’, ‘±15°’, 2];[‘IM6, 65%’, ‘±75°’, 2];[‘IM6, 65%’, ‘±35°‘, 2];

  [‘Mod, 45%’, ‘±65°’, 2];[‘Mod, 45%‘, ‘±35°’, 2];[‘Mod, 45%’, ‘0°’, 2]]

This was the most simplified example of using a greedy algorithm to determine the composite structure. In this case, there were only three calculated cases (when in reality, their number may exceed several dozens), a short pool of materials selection, and a fairly high step of sorting angles. Below (Figure 10), the application of an ordinary local search algorithm is presented.

Based on the graph, it can be seen that each peak is followed by a sharp decline. This algorithm stops searching in the neighborhood and adds a layer as soon as it gets a value that is worse than the previous one. The resulting structure is: 

[[‘AS4, 63%’, ‘±75°’, 8], [‘ IM6, 65%’, ‘±60°’, 4], 

  [‘ AS4, 63%’, ‘±60°’, 2], [‘ AS4, 63%’, 0°, 10]]. 

It has a total thickness of 6 mm which is expected to be worse than the previous one, since the algorithm found only the local optima on each layer. Nevertheless, the result that was obtained is better than the analytical one, especially when considering the real number of the optimized parameters [24].

The final algorithm that was considered in this paper was the Tabu search [25,26,27], which is presented below on Figure 11. More specifically, it is the simplest variation without accelerations or randomizing parameters, with the length of the Tabu list being 10 elements.

Surprisingly, the resulting thickness in this case turned out to be even smaller than in the full search [28,29]. This happened because these are discrete iterative algorithms, and the result of each next step is better than the previous one by some specific amount [30]. The value of the failure criterion (which should be less than 1, but tends to it because the smaller it is, the greater the margin of safety), and in the case of the full search, the algorithm found the best solution. Therefore, the next iteration changed the failure criterion to 0.8263, which rather indicates the imperfection of the first algorithm. The rejection criterion for the latest Tabu search algorithm from this initial value was 0.9562, what explains why this thickness was smaller [31,32].

In order to verify the proposed methods, a comparative analysis was conducted. Table 3 shows the typical loads. Each square parenthesis represents a different observation case, and three different types of loads in each case-they are longitudinal, transverse, and shear loads. The positive load value represents tensile force and the negative represents a compressive force. The loads are in kN.

Table 4 shows the results of the method calculations.

At this point, in each cell, separated by slash: load factor / required thickness / calculation time, for the analytical algorithm there are only thickness and time taken into account. Obviously, the calculation time here has only a relative meaning of comparison [33].

## 4. Conclusions

Based on these research results, heuristic optimization methods provide a very positive result. The works in this field have the potential to explore other approaches to a neighborhood organization and to apply other heuristic methodologies.

The purpose of this work was to demonstrate that the use of heuristic methodologies could significantly help in solving the problems of both design and construction with numerous optimized parameters and with an excessive amount of conditions. 

Another non-obvious advantage of using these methods is the assignment of formulas in a general form and the absence of the need to engage in an analytical analysis, which in the future could accelerate the design process. Using a heuristic approach in the design stage of composite production also makes it possible to reduce the final cost of production. Even though the heuristic optimization is considered inaccurate because, by definition, it does not definitively provide the best solution of all feasible options, this article demonstrates that by using a heuristic approach it is possible to obtain more than acceptable results in the overdetermined problems when an exact solution cannot be obtained in other ways. The main innovation of the proposed approach is the ability to design a composite structure, which is otherwise not possible to find. The ability to compute each single added layer provides a structure with a precise amount of reinforcement to the considered loads, in other words, with no extra unused material. This phenomenon is clearly shown by cases 3, 6, 8, and 10, when the Tabu search designed structures with a load factor that was close to 1 (0.97, 0.99, 1, and 0.98, correspondingly).

However, it is important to proceed with the research, studying more heuristic methodologies and their applications on different structures. It is planned to devote more in-depth attention to a further study on the application of heuristic methods such as annealing simulation, ant colony simulation, as well as genetic algorithms. As it regards different composite structures, the plane lamina study is not considered complete and deeper research is needed to study different failure criteria and operation conditions. Along with planar composite lamina, it is planned to study the application of heuristic optimization on the design methodologies of structures such as composite beams, spars, pivots, and rotation shells.

## Figures and Tables

**Figure 1 materials-16-00495-f001:**
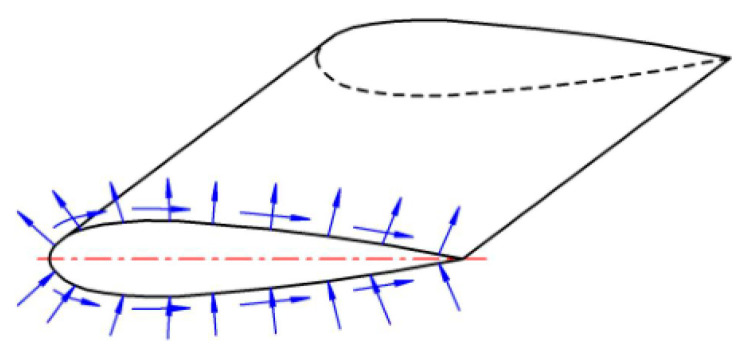
Aerodynamic pressure applied to an aircraft wing.

**Figure 2 materials-16-00495-f002:**
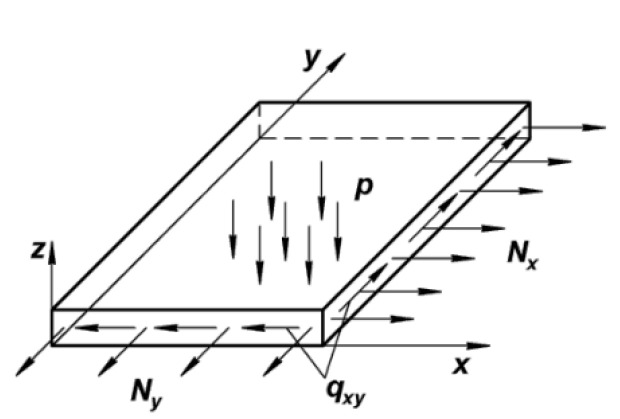
Aerodynamic forces represented in the form of loads that are applied to the conditional section of the aircraft wing skin. N_x_—longitudinal force in the plane, N_y_—transverse force, q_xy_—tangential force flow, and p—pressure, perpendicular to the plane.

**Figure 3 materials-16-00495-f003:**
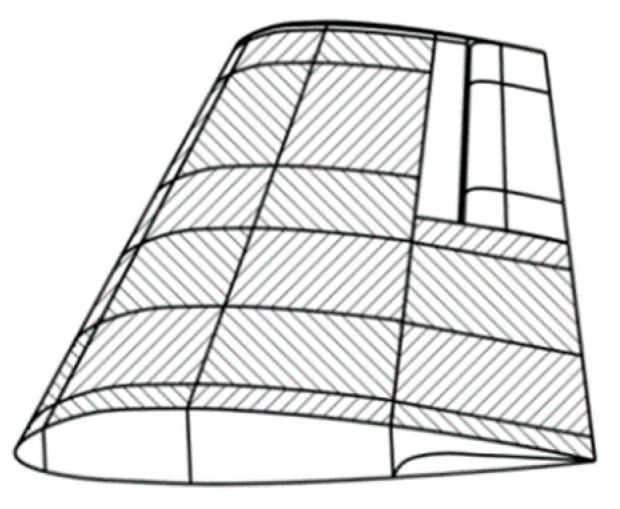
Contingent sections of the aircraft wing between ribs and stringers.

**Figure 4 materials-16-00495-f004:**
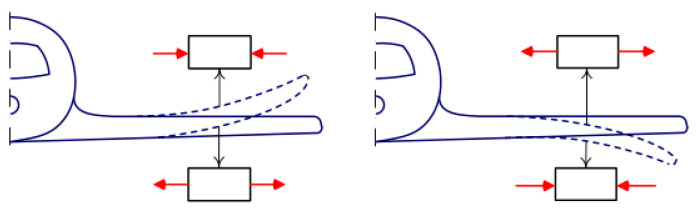
Deformations due to aerodynamic forces applied to an aircraft wing.

**Figure 5 materials-16-00495-f005:**
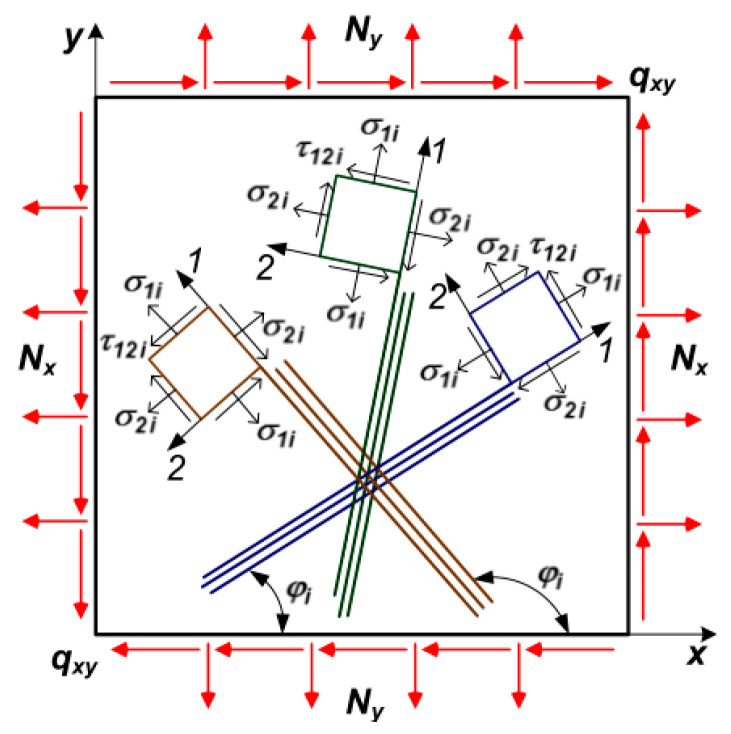
Structure of composite lamina.

**Figure 6 materials-16-00495-f006:**
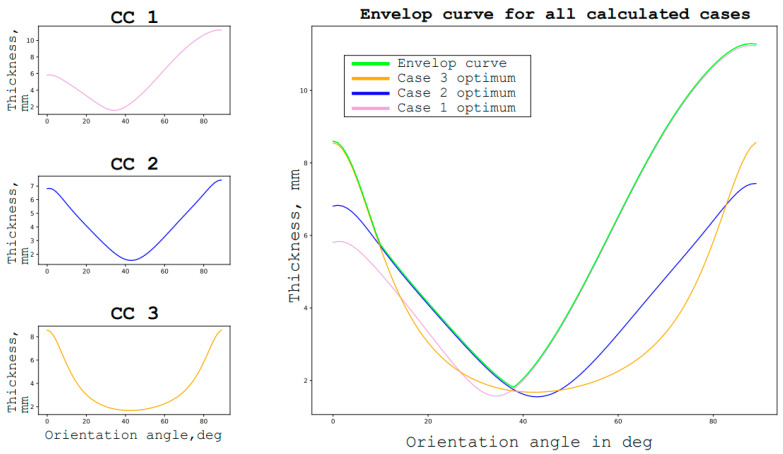
Strength failure coefficients for the different layer orientations for various calculative cases for a specific case study.

**Figure 7 materials-16-00495-f007:**
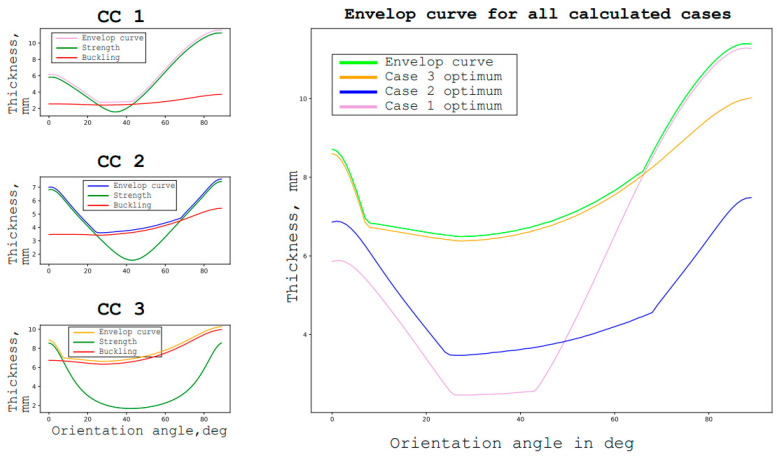
Strength and buckling failure coefficient for different layer orientations for various observed cases for a specific case study.

**Figure 8 materials-16-00495-f008:**
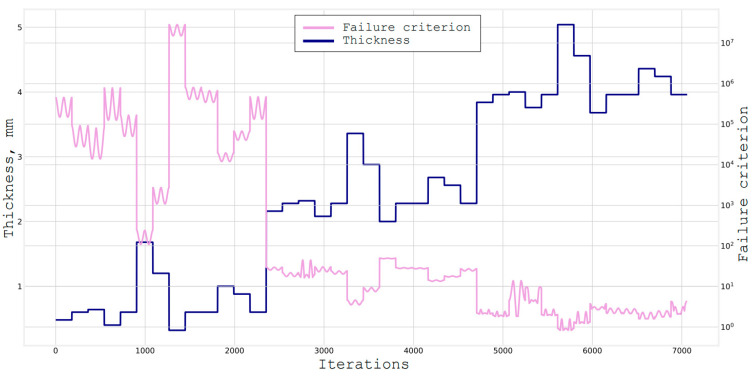
Full search through materials and angles with step of 1°, and a thickness = 5.88 mm 485 s.

**Figure 9 materials-16-00495-f009:**
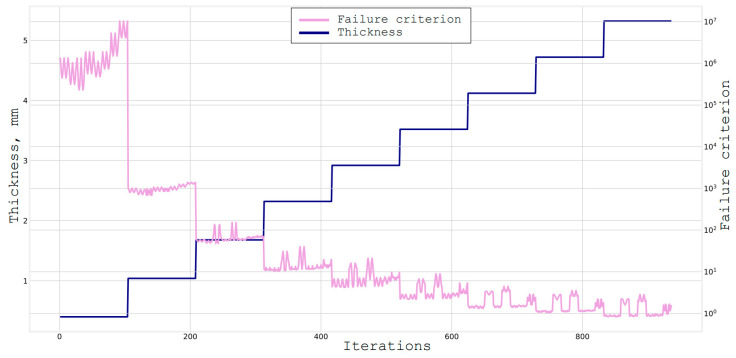
Full search through the unit materials and angles with step of 15 deg, and a thickness = 4.92 mm 412 sec.

**Figure 10 materials-16-00495-f010:**
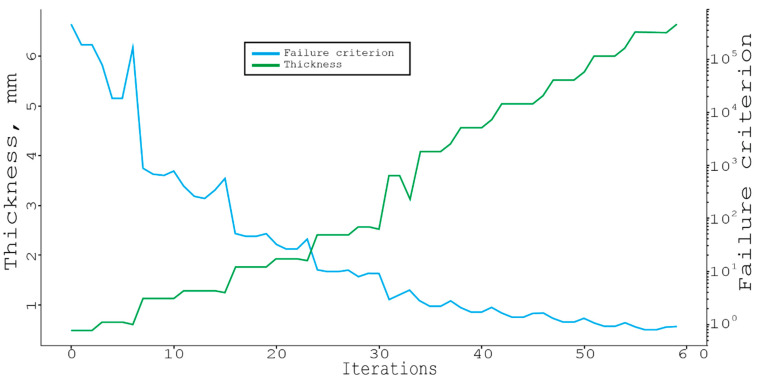
Local search through materials and angles with step of 15 deg, and a thickness = 6 mm 49.6 sec, failure criterion = 0.84.

**Figure 11 materials-16-00495-f011:**
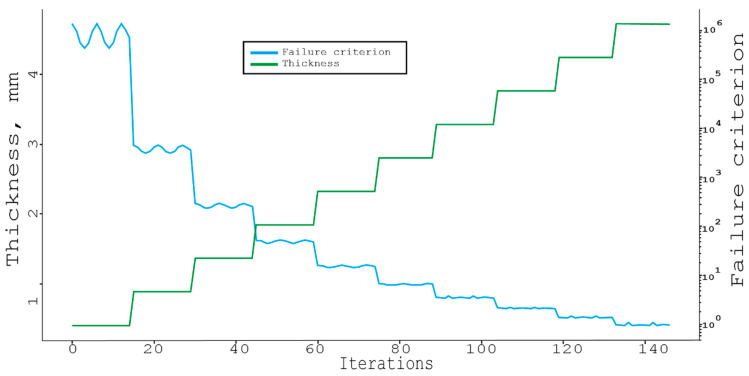
Tabu search through materials and angles with the step of 15 deg and a thickness = 4.8 mm 70.64 s, failure criterion = 0.92.

**Table 1 materials-16-00495-t001:** Common carbon fiber-based and organic fiber materials (case 1-3).

Load Direction	Case 1	Case 2	Case 3
N_x_	−600	0	−300
N_y_	0	−500	−400
q_xy_	50	0	100

**Table 2 materials-16-00495-t002:** Common carbon fiber-based and organic fiber materials.

No	1	2	3	4	5	6	7	8	9
Type	Carbon							Organic	
Fiber	AS4, 63%	IM6, 65%	ModI, 45%	GY70, 57%	AS4, 58%	AGP 3705 5H, 62%	CF 0604, 55%	Kevlar 49, 60%	K120, 45%
Matrix	epoxy 3501-6	epoxy sc1081	WRD 9371	epoxy 934	PEEK APC2	epoxy 3501-6	epoxy 934	epoxy M10.2	epoxy M10.2
E_1_, GPa	147	177	216	294	131	77	65.6	80	29
E_2_, GPa	10.3	10.8	5	6.4	8.7	75	60.3	5.5	29
G_12_, GPa	7	7.6	4.5	4.9	5	6.5	3.98	2.2	1.8
µ_12_	0.27	0.27	0.25	0.23	0.28	0.06	0.04	0.34	0.05
F_1t_, MPa	2280	2860	807	589	2060	963	927	1400	369
F_1c_, MPa	1725	1875	665	491	1080	900	729	335	129
F_2t_, MPa	57	49	15	29.4	78	856	874	30	369
F_2c_, MPa	228	246	71	98.1	196	900	620	158	129
F_12_, MPa	76	83	22	49.1	157	71	133	49	113
ρ, kg/m^3^	1580	1600	1540	1590	1570	1600	1560	1380	1380
δ^0^, mm	0.12	0.15	0.16	0.1	0.15	0.42	0.3	0.15	0.22

**Table 3 materials-16-00495-t003:** Pool of typical loads.

No	Loads				
1	65, −255, −670	0, 0, 780	−335, −245, 195	−465, −525, 115	265, 0, 0
2	0, 665, 605	-150, 0, 475	0, -800, −425	0, −670, 0	0, 415, −650
3	−355, 730, −770	−330, −740, −695	0, 0, 0	−600, −275, 775	−640, 385, 525
4	−730, 0, 485	70, 600, 710	0, 205, 0	430, −200, 0	620, 665, 585
5	−335, 735, −630	−605, −690, −645	0, 250, 0	0, 0, −145	0, −60, 755
6	780, −10, 0	755, 0, 215	−460, −775, 0	315, −710, −510	0, 0, −180
7	0, 460, 135	0, 0, 480	25, 0, 0	0, 755, 245	0, 0, −745
8	0, 0, 360	580, 430, −225	485, 0, 635	−325, 500, 0	−295, 575, −570
9	0, −665, 550	−740, −770, 0	705, 710, 0	0, −575, −35	640, 0, 0
10	410, 115, 55	0, 0, −15	−450, 0, 0	170, −485, −245	0, −195, 0

**Table 4 materials-16-00495-t004:** Results of the calculations.

No	Analytic	Full Greedy	Local Greedy	Tabu Search
1	9.53 / 4.1	0.96 / 5.52 / 942	0.85 / 5.28 / 64	0.84 / 5.28 / 158
2	8.91 / 3.4	0.98 / 6.08 / 1271	0.92 / 5.04 / 47	0.8 / 5.28 / 154
3	11.08 / 4.8	0.96 / 8.48 / 3271	0.99 / 6.0 / 73	0.97 / 5.76 / 205
4	10.82 / 2.9	1.13 / 6.68 / 1656	0.97 / 8.16 / 131	0.76 / 5.76 / 199
5	12.21 / 3.9	1.04 / 7.28 / 2116	0.87 / 5.88 / 82	0.73 / 6.24 / 256
6	12.01 / 3.4	0.96 / 6.64 / 1649	0.74 / 8.16 / 220	0.99 / 7.2 / 375
7	6.68 / 2.7	0.97 / 5.48 / 938	0.69 / 3.96 / 24	0.78 / 3.84 / 65
8	6.07 / 2.7	0.94 / 6.68 / 1654	0.98 / 12.96 / 442	1.0 / 5.28 / 145
9	12.84 / 3.8	1.09 / 6.08 / 1270	0.93 / 7.68 / 174	0.74 / 7.2 / 379
10	8.6 / 3.1	0.73 / 4.24 / 473	0.95 / 4.92 / 41	0.98 / 4.32 / 87.4

## Data Availability

Not applicable.

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
