# Peer review of "Composite Lamina Model Design with the Use of Heuristic Optimization"

_materials, 2023, doi:10.3390/ma16020495_

Round 1

Reviewer 1 Report

The paper presents a different approach to assessing the optimal use of composite materials in an aircraft wing by means of heuristic optimization. The methods are well-developed and the results are interesting, however, the paper often uses informal language, and also could use an English editing service to improve its quality and remove some typos.

Author Response

Thank you for the review, we have attached the answers to the questions.

Reviewer 2 Report

The paper presents a study on composite lamina model design with the use of heuristic optimization

The following recommendations are proposed:

·         Please, improve the introduction section. It misses a comprehensive literature review.

·         Please, provide a flow chart of the paper organization.

·         Overall, English needs to be double-checked for typos.

·         The conclusions section can be better organized.

Main Concern:

What is the innovation that this paper brings into scientific knowledge?

Author Response

(The authors gave the same response as above.)

Reviewer 3 Report

The present article represents a way to find the optimal solution to under-determined problem by heuristic optimization methods using the example of designing a composite wing skin of an aircraft. As the authors mentioned that through a heuristic approach, which is not very accurate for optimization, it is possible to obtain more than acceptable results in overdetermined problems when an exact solution cannot be obtained. The paper can be published if the authors respond to the following queries:

 1. The abstract only explains what has done in the paper in few lines and 70% of the abstract explains the importance of optimization through numerical method in engineering problems. I suggest authors should include more details of their solution method and conclusions in the abstract section.

2. Throughout the paper, there are unnecessary hyphens “-” between words which can be removed.

3. Can the authors point out to a particular graph from a previous work which has been reproduced by this method?

4. Line 180, Page 6, correct subsection title “2.4. Streng Buckling failure criterion” to “2.4. Strength Buckling failure criterion”

Author Response

(The authors gave the same response as above.)

Reviewer 4 Report

·       The problem in the introduction can be defined specifically.

·       The Figures 4, 6 captions can be better with description. 

·       The equations should be explained thoroughly. 

·       The section 2.4 heading should be correct.

·       Please elaborate on the influence of thickness on the failure criterion. 

·       The conclusion may exclude the Tables 3 and 4.  These can be included in the results section.  The conclusion should be rewritten.       

Author Response

(The authors gave the same response as above.)

Round 2

Reviewer 2 Report

Comments addressed